# Polysaccharide Paint Binding Media at Two Pharaonic Settlements in Nubia

Kate Fulcher [1,*], Neal Spencer [2], Julia Budka [3] and Rebecca J. Stacey [1]

[1] Department of Scientific Research, British Museum, London WC1B 3DG, UK; rstacey@britishmuseum.org
[2] The Fitzwilliam Museum, University of Cambridge, Cambridge CB2 1RB, UK; nas1003@cam.ac.uk
[3] Egyptian Archaeology and Art History, LMU Munich, 80333 Munich, Germany; julia.budka@aegyp.fak12.uni-muenchen.de
[*] Correspondence: kfulcher@britishmuseum.org

**Abstract:** Paints and plasters from two pharaonic settlement sites in Nubia (northern Sudan) were analysed to investigate the presence and origin of organic binding materials. The town of Sai was founded around the time of the pharaonic conquest of Kush (Upper Nubia) around 1500 BC, with Amara West created as a new centre for the pharaonic administration of the region around 1300 BC. Recent fieldwork at both sites yielded examples of paint palettes, including several from houses. These provide a different economic and social context to funerary contexts upon which most previous research has been conducted, making this study the first to report on binding media for vernacular architecture in the Nile Valley. It is also the first study of binding media from Nubia. Gas chromatography mass spectrometry (GC-MS) analysis of methanolysed and silylated paint and plaster samples revealed a range of monosaccharides present in eight of the seventeen samples from Amara West, and in six of the seven samples from Sai. Interpretation of the data was supported by field collection and study of locally available botanical gums. The results indicate that mixtures of gums were in use as a pigment binder at both sites during the mid- to late-second millennium BC. The possibility that some of these plant gums could have been imported from the Mediterranean is also posited.

**Keywords:** archaeology; Egypt; Nubia; pigment; botany; gums; colonialism; urbanism; technology

## 1. Introduction

### 1.1. Background and Aims

The study of ancient Egyptian paint and painting technology has focused more on the pigment components than the binding media [1,2], as pigments provide the most visible property of paint: colour. Moreover, they can be characterised by the minimally destructive analytical methods that are preferred for museum artefacts, and most are more resistant to degradation than organic binding media. The limited published research on binding media indicate the predominance of polysaccharide gums in ancient Egyptian paint binding, although protein binders have sometimes been reported [3–6].

To date, no examples of binding media from ancient Nubia have been published. This study focuses on paint materials from two sites in northern Sudan—Sai and Amara West—dating to a period of Pharaonic rule in Nubia (c. 1500–1070 BC) and seeks to examine how paint binding technology compares with that in use in contemporaneous Egypt. The excavated sites provide particular opportunities for such research as finds of bulk paint materials offer greater potential to identify binding media through sampling, and finds from domestic contexts may reveal different perspectives than the information gleaned from elite contexts such as burial goods or painted tomb decoration.

### 1.2. The Sites

The two sites considered in this study, founded by the Egyptian state to control and administer the region and facilitate the extraction of resources, provide opportunities to investigate aspects of lived experience, cultural entanglement, production and resource management in the context of pharaonic colonialism in northern Sudan during the New Kingdom.

The town on Sai Island, located on a large island approximately halfway between the Second and Third Cataracts, was founded during, or soon after, the initial conquest of the region in the early 18th Dynasty (c. 1500 BC) [7], and was occupied throughout the remainder of the period of Egyptian rule. Two sectors within the walled town, SAV1 West and SAV1 East, were excavated by the AcrossBorders project [7,8] and yielded ceramic sherds containing paint [9]. All of the samples used in the current study were excavated in SAV1 West, which featured a street and domestic residences. The samples were found near the town enclosure wall, in deposits rich with ceramics, which range in date from the early to late 18th Dynasty.

The town of Amara West, 20 km downstream of Sai (Figure 1), was founded around two centuries later than Sai, in the reign of Seti I (c. 1300 BC) and was occupied until the late 11th century BC, though burials in the cemetery continued into the 8th century BC [10]. Fieldwork undertaken by the British Museum's Amara West Research Project (2008–2019) uncovered a large quantity of materials relating to painting: small lumps of natural and human-made pigments, ceramic sherds re-used as palettes to mix paints, grindstones with traces of pigment, painted walls and features within houses and the remains of painted wooden coffins placed in the tombs [11]. The largest concentration of material was found in area E13, in a facility (E13.14) used for both storage and production activities, then partly re-purposed as workshops (E13.29, E13.31). These areas were subsequently converted into houses by the insertion of additional walls and doorways (E13.7, E13.3, E13.6), around the same time that a 'suburb' developed outside the town walls, with notably larger dwellings (e.g., house D12.5).

The investigation of the binding media in the paint materials reported here was carried out alongside an extensive multi-analytical study of the pigments used at both sites. Some 300 pigments from Amara West were investigated [11] and twenty-six from Sai [9]. At both sites, the analyses identified yellow and red ochres (iron oxides with natural admixtures); whites made from gypsum, anhydrite (dehydrated gypsum), calcite, or, more rarely, huntite; blues were Egyptian blue, except for four palettes at Amara West which were identified as riebeckite [12]. At Amara West, bitumen and carbon blacks were identified [13] and a few greens: copper chloride from a grindstone, green earth chlorite on some palettes and a single example of mixed Egyptian blue and yellow ochre [12]. Mixtures with white, especially with ochres, were observed at both sites [9,11].

### 1.3. Approach

Study of the binding media was undertaken using molecular analysis by gas chromatography mass spectrometry (GC-MS) to detect polysaccharide binding media in the paints and plasters and identify the plants from which they were sourced. Previous infrared spectroscopy analysis did not suggest the presence of proteins [9,11], and all of the paints presented a powdery appearance, rather than glossy [14], so the presence of proteins was not investigated in this study.

Plant gums are predominately composed of polysaccharides—polymers comprising monosaccharides bound together by glycosidic linkages. Cleavage of these linkages by hydrolysis allows the monosaccharides and corresponding uronic acids to be analysed by GC-MS and identified [15]. To some extent the original plant gum used can be determined from the types of monosaccharides present and their relative quantities. Published analyses of plant gums report the presence of the monosaccharides arabinose, fucose, xylose, mannose, rhamnose, galactose and glucose in varying quantities [15–18]. The complex polysaccharides in plant gums can vary from tree to tree, even among the same species.

Sudanese *Acacia*, for example, has been reported as having both significant and negligible amounts of rhamnose [19,20]. To investigate this phenomenon in the relevant geographical area, modern *Acacia nilotica* gums from near the site of Amara West were analysed.

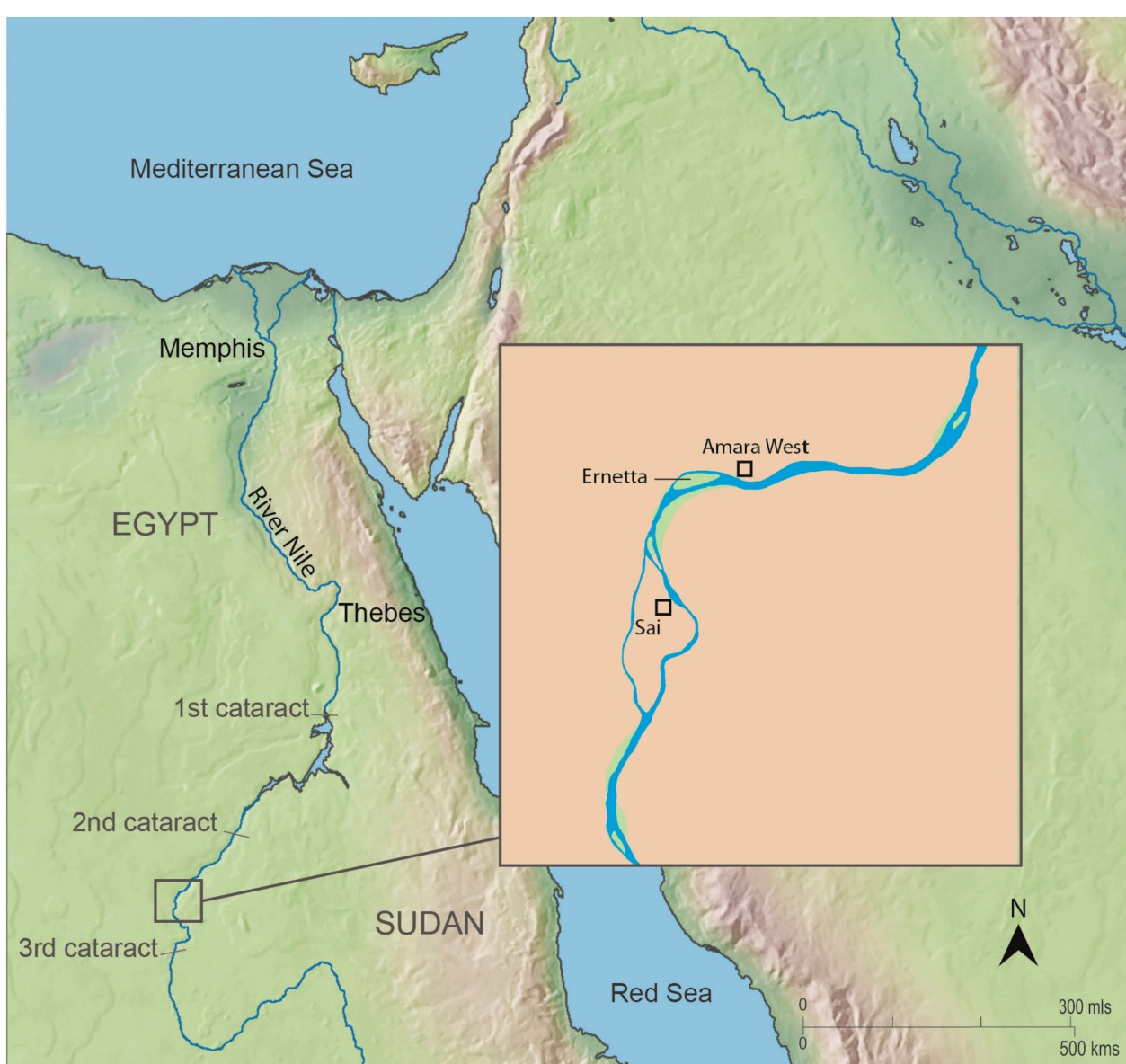

**Figure 1.** Map of a portion of Egypt and Sudan showing the location of Amara West, Sai Island and Ernetta Island. Map drawn by Anthony Simpson and Kate Fulcher.

## 2. Samples

Seven samples from Sai, and seventeen samples from Amara West, were studied (Table 1), selected for their range of colours and volume of material available. All the samples were taken from paint on ceramic palettes or from pieces of wall plaster, which allowed for large volume sampling (0.5 to 1 g) thereby maximizing the potential for recovery of organic binder. Reference samples of *Acacia* (gum Arabic), *Prunus* (plum), and *Astragalus* (tragacanth) were taken from the British Museum Reference Collection.

**Table 1.** List of samples analysed. The description includes the colour of the paint or plaster and the mineral identification [9,11]. For Sai, the provenance designates the square and stratigraphic unit. For Amara West, provenance designates the grid/building/room followed by the stratigraphic unit [21].

| Sample Number | Find Number | Provenance | Date | Description |
|---|---|---|---|---|
| **Sai** | | | | |
| Sai130 | SAV1W 0550/2015 | SAV1W, Sq. 1S, SU 679 | c. 1400 BC | Palette with red ochre |
| Sai133 | SAV1W 0650/2015 | SAV1W, Sq. 1S, SU 698 | c. 1400–1300 BC | Palette with huntite |
| Sai150 | SAV1W 0230/2015 | SAV1W, Sq. 1S, SU 630 | c. 1540–1300 BC | Palette with Egyptian blue |
| Sai153 | SAV1W P045 | SAV1W, Sq. 1, SU 537 | c. 1400–1300 BC | Palette with yellow ochre, gypsum, calcite |
| Sai154 | SAV1W P048 | SAV1W, Sq. 1, SU 507 | c. 1540–1300 BC | Palette with yellow ochre, gypsum, calcite |
| Sai155 | SAV1W P051 | SAV1W, Sq. 1, SU 585 | c. 1540–1400 BC | Palette with anhydrite |
| Sai163 | SAV1W P069.1 | SAV1W, Sq. 1, SU 507 | c. 1540–1300 BC | Palette with red and yellow ochre, gypsum |
| **Amara West** | | | | |
| AW119 | F17310 | E13.29.1 ((5230) | c. 1200 BC | Palette with bitumen |
| AW121 | F17311 | E13.29.4 (5243) | c. 1180 BC | Palette with bitumen, carbon |
| AW122 | F6190 | E13.31.2 (5332) | c. 1200 BC | Palette with gypsum, calcite |
| AW127 | F6170 | E13.6.4 (5341) | c. 1200 BC | Palette with Egyptian blue, gypsum |
| AW129 | F6079 | E13.29.4 (5222) | c. 1180 BC | Palette with yellow ochre, calcite |
| AW130 | F6147 | E13.31.1 (5325) | c. 1180 BC | Palette with red ochre |
| AW132y | F6147 | E13.31.1 (5325) | c. 1180 BC | Palette with yellow ochre, anhydrite |
| AW132b | F6147 | E13.31.1 (5325) | c. 1180 BC | Palette with Egyptian blue |
| AW133 | F6147 | E13.31.1 (5325) | c. 1180 BC | Palette with anhydrite |
| AW139 | F6281 | E13.29.3 (5246) | c. 1180 BC | Palette with carbon |
| AW141 | F6408 | E13.31.2 (5334) | c. 1200 BC | Palette red ochre, calcite |
| AW256 | | E13.7.6 (4566) | c. 1200 BC | Gypsum wall plaster |
| AW259 | | E13.3.24 (4068) | c. 1200–1180 BC | Gypsum wall plaster |
| AW323 | F7278 | D12.5.12 (2538) | c. 1200–1100 BC | Palette with yellow ochre |
| AW435 | F6264 | E13.29.2 (5261) | c. 1180 BC | Palette with red ochre, calcite |
| AW440 | F6493 | E13.14.1 (5361) | c. 1190 BC | Palette yellow ochre |
| AW445 | F6446 | E13.31.3 (5339) | c. 1200 BC | Palette with gypsum |

Modern gum samples were collected from eight *Acacia* trees on Ernetta Island (20°49′ N, 30°21′ E), located in the Nile between Sai and Amara West (Figure 1), in February 2017. Trees extrude gum in response to damage, which may be accidental or deliberately inflicted to stimulate gum production for later collection [22]. In this case, gum was harvested from previously damaged areas. Targeted trees were those with the spiny leaves, yellow pom-pom flowers and circular seed pods (Figure 2) that identified them as *Acacia nilotica* (also known as *Vachellia nilotica*) [23], a species whose seeds were found in houses at both sites [24,25]. Five samples were amber-coloured fresh gum, some still sticky when picked; three samples were dried and blackened (Figure 3).

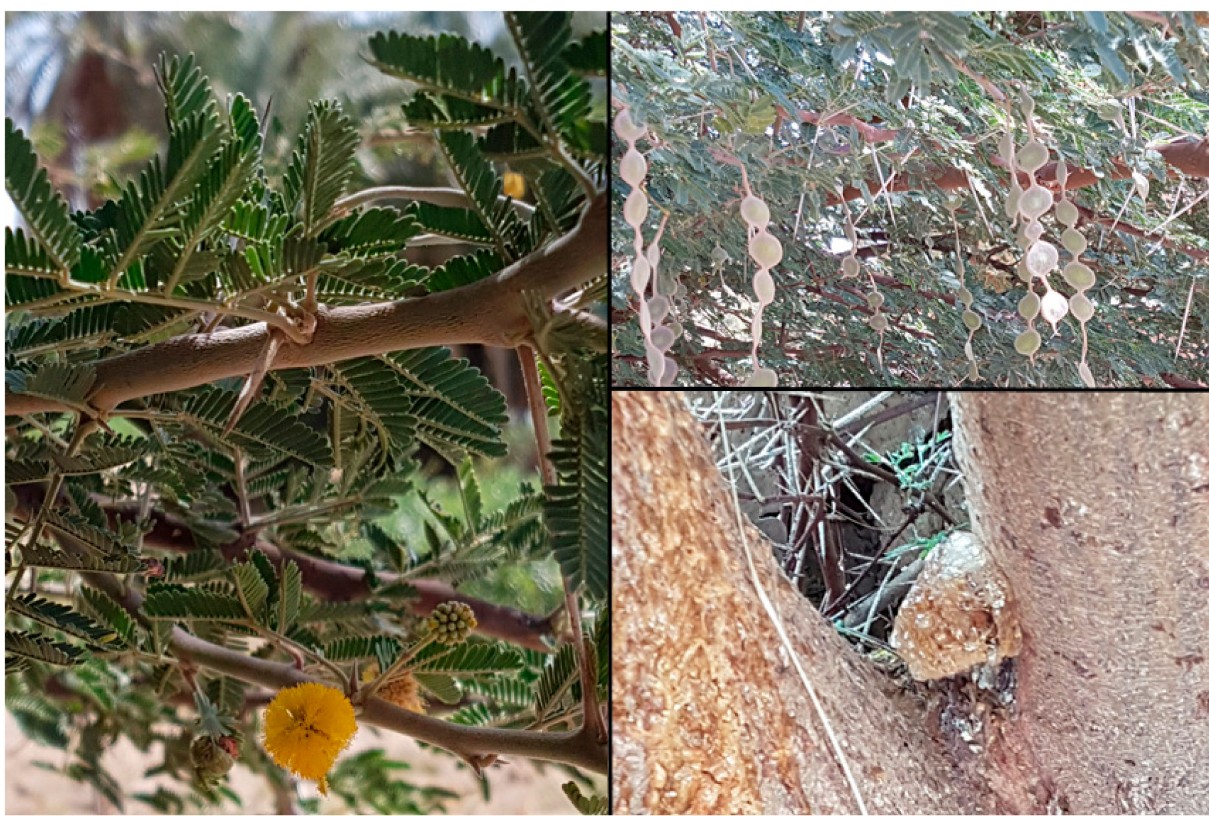

**Figure 2.** Trees from which gum samples were taken on Ernetta Island. **Left**: leaves and flowers indicating species *Acacia nilotica*; **Right, above**: seed pods; **Right, below**: lump of gum on tree. Photographs: Kate Fulcher.

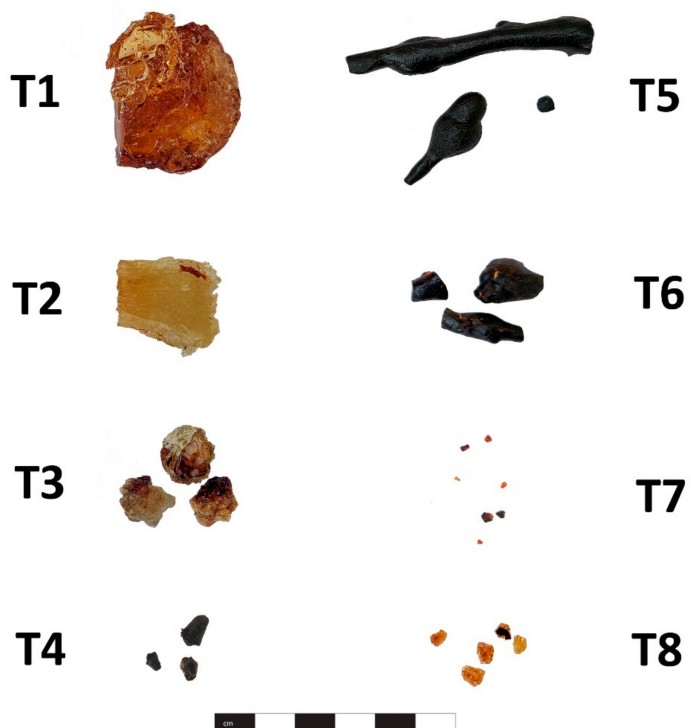

**Figure 3.** Samples T1 to T8 taken from *Acacia nilotica* trees on Ernetta Island.

## 3. Method

All samples were analysed at the British Museum. The analytical method followed the standard procedure used at the British Museum for the preparation of polysaccharide samples for GC-MS analysis of neutral sugars and uronic acids, which is based on the method published by Bleton et al. [15]. Samples and reference samples were powdered before being hydrolysed by the addition of 500 µL of 0.5 M hydrochloric acid, heated at 80 °C for 20 h. The solution was decanted and dried under nitrogen. The dry hydrolysed extracts were derivatised by the addition of 300 µL Sigma-Sil A (1:3:9 ratio of trimethylchlorosilane (TMCS), hexamethyldisilazane (HMDS) and pyridine), and heated at 80 °C for 2 h. Samples were dried under nitrogen and dissolved in 100 µL hexane in preparation for injection into the GC-MS instrument. Blanks were prepared alongside the samples using the same method.

Derivatised samples were separated using an Agilent HP5-MS column (30 m × 0.25 mm, 0.25 µm film thickness with 1 m × 0.53 mm retention gap) with splitless injection at 300 °C and 10.1 psi and a purge time of 0.5 min. The carrier gas was helium with a flow at 1.5 mL/min. The oven was set at 40 °C to 130 °C at 9 °C/min, then to 290 °C at 2 °C/min, with the final temperature held for 10 min. Two instruments were used. Samples from Amara West and Ernetta Island were analysed using an Agilent 5973 MSD operating in the electron impact (EI) mode at 70 eV and scanning over the range $m/z$ 50 to 550. Samples from Sai were analysed using an Agilent 5977 MSD operating in the electron impact (EI) mode at 70 eV and scanning over the range $m/z$ 50 to 800. Total ion chromatograms were extracted for ions $m/z$ 217 and 204 to view any furanoside and pyranoside compounds in the sample [15,26], and data were interpreted using the NIST database version 2.3, reference samples (run on both MSDs), and examples in the literature (Table 2). Interpretation requires a combination of retention time and spectral features, since the method separates multiple epimers with near identical spectra.

**Table 2.** Results of gum analysis of paint samples from Amara West and Sai, the samples taken from trees on Ernetta Island, and examples of gum analysis from the literature for comparison. *Acacia* is also referred to as gum Arabic; *Acacia arabica* is a synonym for *Acacia nilotica* [27].

| | Arabinose | Rhamnose | Galactose | Glucose | Mannose | Xylose | Fucose | Glucuronic Acid | Galacturonic Acid |
|---|---|---|---|---|---|---|---|---|---|
| **Samples from Sai** | | | | | | | | | |
| Sai130 | + | ++ | ++ | ++ | ++ | ++ | + | | + |
| Sai133 | ++ | ++ | + | + | + | ++ | ++ | | + |
| Sai150 | | ++ | + | | | + | ++ | | ++ |
| Sai153 | ++ | + | ++ | ++ | ++ | + | + | + | ++ |
| Sai154 | + | ++ | ++ | ++ | + | + | ++ | + | + |
| Sai155 | | ++ | + | + | + | | + | | |
| **Samples from Amara West** | | | | | | | | | |
| AW121 | | | ++ | | + | | + | | |
| AW129 | + | | + | + | + | | | | (+) |
| AW132b | + | | + | | + | (+) | | | |
| AW139 | + | | + | | + | | | | |
| AW256 | + | + | + | ++ | + | + | + | | + |
| AW323 | | | + | | + | | | | |
| AW435 | + | + | ++ | ++ | + | + | + | | |
| AW445 | + | ++ | + | ++ | + | + | + | + | + |
| **Samples from Ernetta Island** | | | | | | | | | |
| T1–3, T7–8 (amber) | ++ | + | ++ | | | | | + | + |
| T4–6 (black) | + | | + | | | | | | |

**Table 2.** *Cont.*

| | Arabinose | Rhamnose | Galactose | Glucose | Mannose | Xylose | Fucose | Glucuronic Acid | Galacturonic Acid |
|---|---|---|---|---|---|---|---|---|---|
| **Modern samples from the British Museum Reference Collection** | | | | | | | | | |
| *Acacia arabica* [1] | ++ | ++ | ++ | | | | | + | + |
| Gum tragacanth [2] | ++ | + | + | | | ++ | + | + | + |
| *Prunus* gum [3] | ++ | + | ++ | + | + | + | | + | + |
| **Examples from literature (modern)** | | | | | | | | | |
| *Acacia arabica* [a] | ++ | + | ++ | | | | | + | + |
| *Acacia arabica* [b] | + | + | + | | | | | + | |
| *Acacia nilotica* [c] | ++ | (+) | ++ | | | | | + | |
| *Acacia nilotica* [d] | ++ | + | ++ | | | | | + | |
| *Acacia Senegal* [e] | ++ | + | ++ | | | | | + | |
| Gum tragacanth [f] | ++ | (+) | + | + | | ++ | + | | ++ |
| Gum tragacanth [b] | + | | + | + | + | + | + | | + |
| Apricot gum [g] | ++ | | ++ | | + | | | + | |
| Cherry gum [a] | ++ | | + | | + | + | | + | |

[1] REFC 66,909: commercial acacia powder; [2] REFC 66,931; [3] REFC 138-Y: collected from UK orchard tree. [a] [28]; [b] [29]; [c] [30]; [d] [20] (Sudanese sample); [e] [31]; [f] [32]; [g] [33]. ++ high proportion; + lower proportion; (+) traces.

## 4. Results

Six of the seven samples from Sai, and eight of the 17 paint samples from Amara West, contained identifiable monosaccharides. Details of the molecular profiles for all samples are summarised in Table 2 and example chromatograms are presented in Figure 4. Minor differences in retention times in the data presented are due to the use of two separate instruments for AW435 and Sai154 (see Method). Samples AW435 and T1 were analysed using the same instrument, but before and after a shut-down period in 2020 (due to COVID-19 lockdown), leading to minor retention time shifts resulting from instrument maintenance.

The samples of amber-coloured gum (T1–3 and T7–8) from *Acacia nilotica* trees on Ernetta Island all produced very similar chromatograms, with very large peaks for arabinose, smaller peaks for galactose, and minor peaks for rhamnose and the uronic acids (Figure 5). The black-coloured gum samples contained minor amounts of arabinose and galactose, but their major constituent was an inositol, possibly D-pinitol (Figures 6 and 7).

Table 2 also lists the monosaccharides identified in the reference samples and analyses of plant gums reported in the literature.

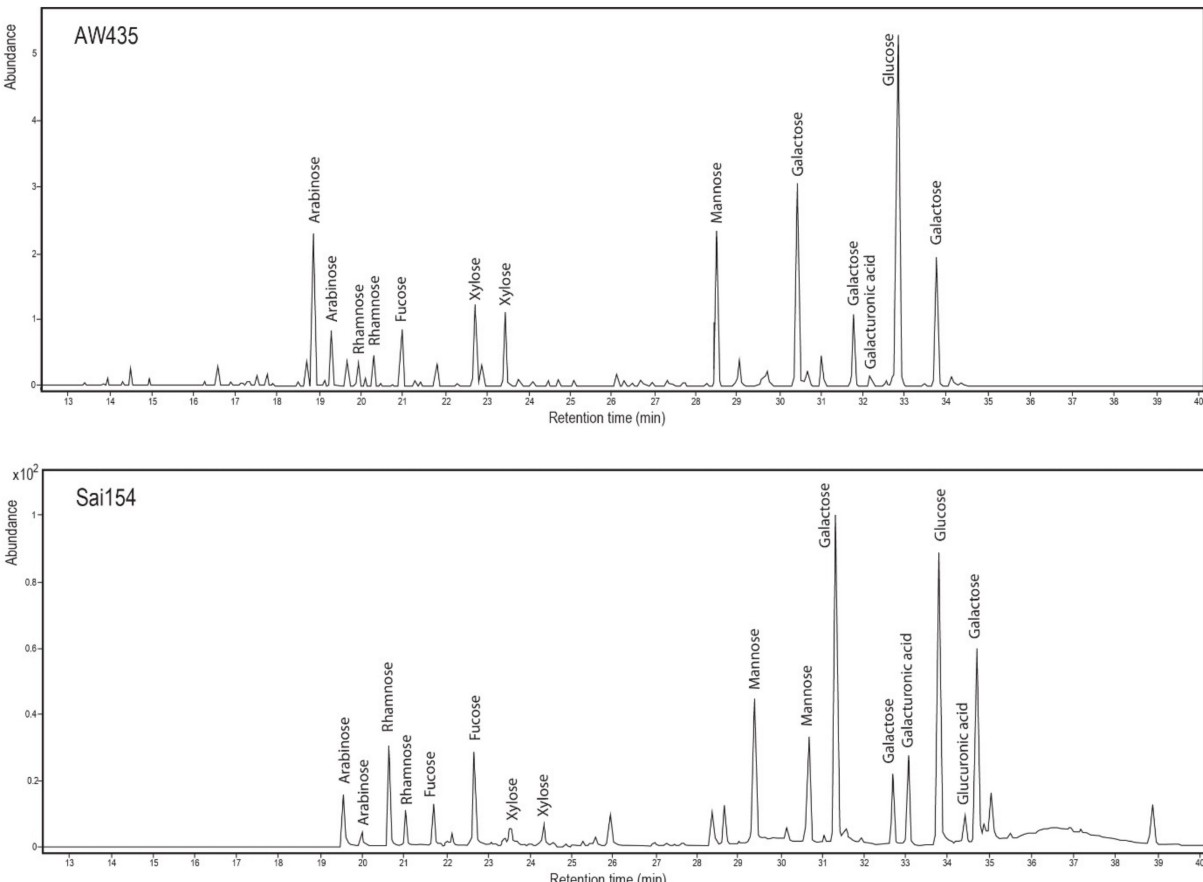

**Figure 4.** Partial (13 to 40 min) extracted ion chromatograms (*m/z* 204 & 217) for obtained from samples AW435 (palette containing red ochre and calcite from E13.7.12) and Sai154 (palette containing yellow ochre, gypsum and calcite from SAV1W, Sq. 1, SU 507).

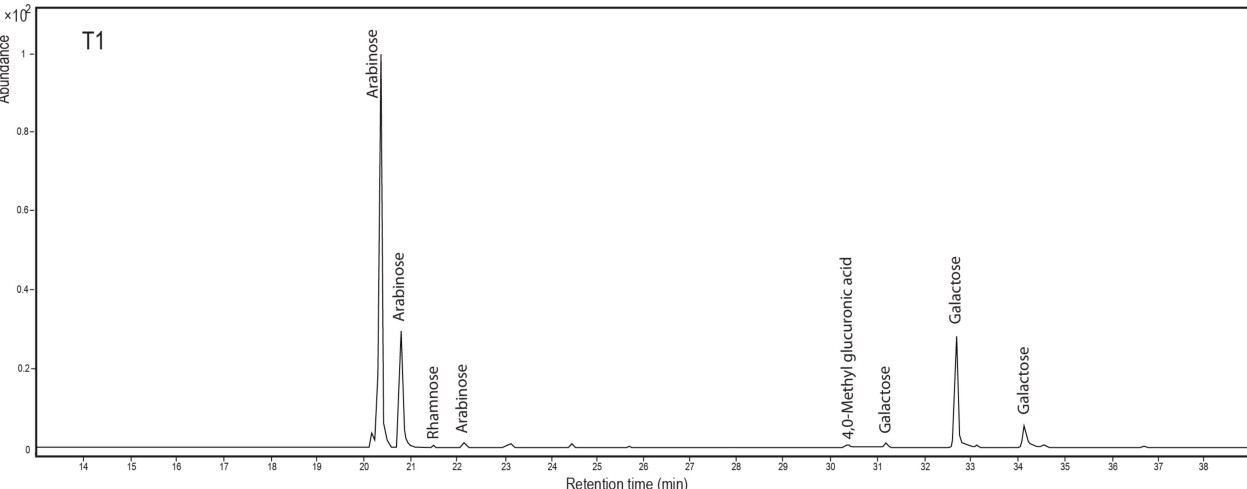

**Figure 5.** Partial (13 to 38 min) extracted ion chromatogram (*m/z* 204 & 217) for Acacia sample (amber coloured) T1 from Ernetta.

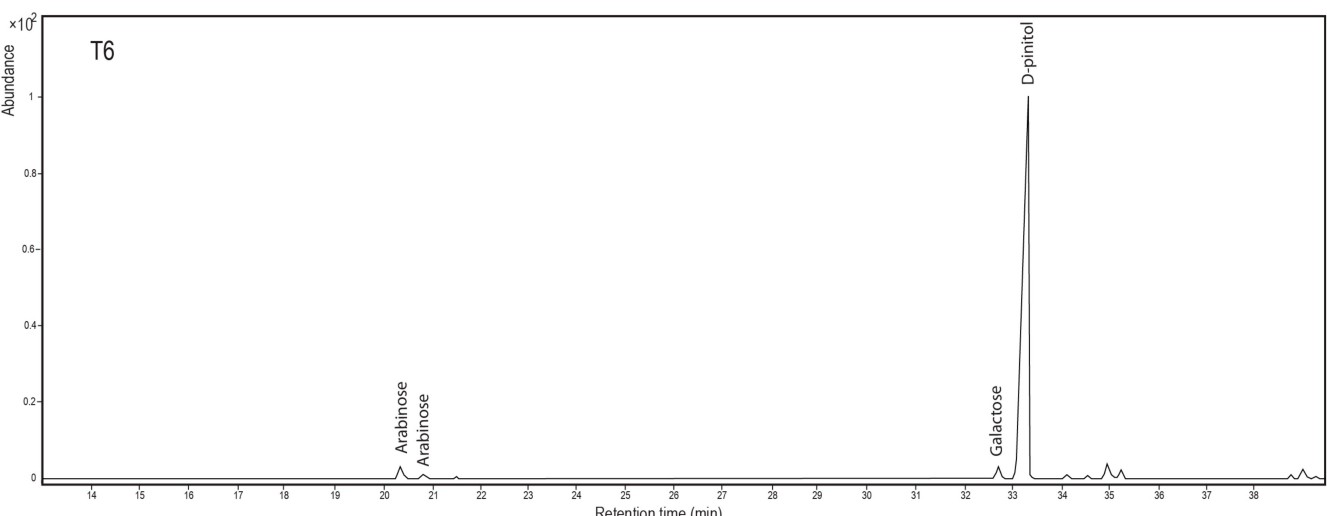

**Figure 6.** Partial (13 to 38 min) extracted ion chromatogram (*m/z* 204 & 217) for blackened Acacia sample T6 from Ernetta Island.

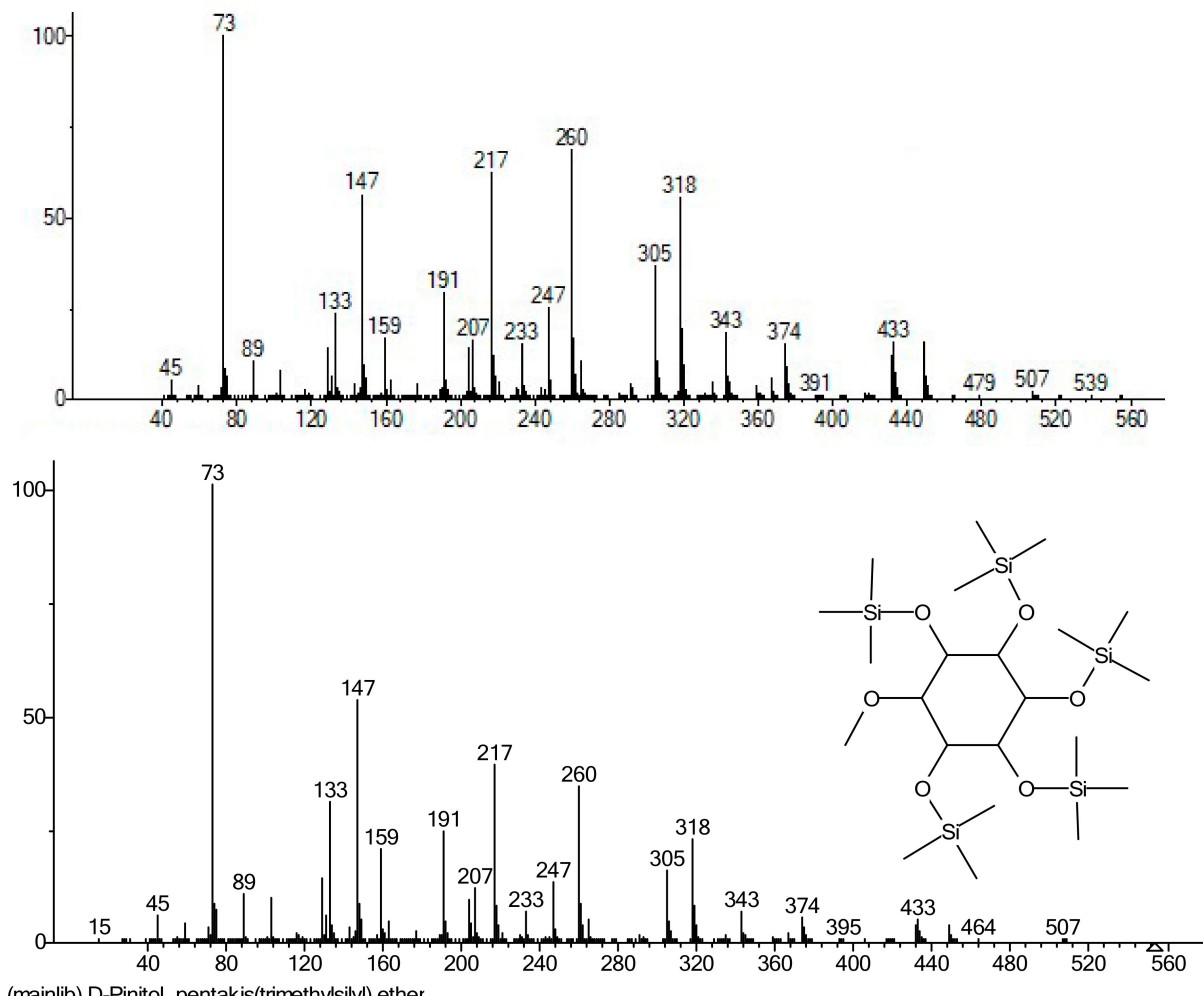

(mainlib) D-Pinitol, pentakis(trimethylsilyl) ether

**Figure 7. Above**: Mass spectrum of the most abundant component (peak at 33.3 min) in sample T6, blackened Acacia gum from Ernetta Island. **Below**: Mass spectrum for D-Pinitol, pentakis (trimethylsilyl) ether from NIST/EPA/NIH Mass Spectral Library (NIST 17) and NIST Mass Spectral Search Program (Version 2.3) from the NIST Mass Spectrometry Data Center, Gaithersburg, MD, USA.

## 5. Discussion

Interpretation of gum spectra from ancient samples is not straightforward, as degradation over time leads to change in the chemical composition. In addition, as Table 2 illustrates, variation in monosaccharides occurs even across reference spectra of the same species in fresh material. Therefore, any identification of specific species must be tentative at best. Significant changes in climate and plant use in Nubia since ancient times make comparisons to modern flora difficult [34]. The range of gums thought to have been available to ancient Egyptians originate from species in the genera *Acacia* (also known as gum Arabic) and *Astragalus* (e.g., tragacanth), and fruit gums [35]. Other sugary substances would have included honey and fruit juices.

The amber-coloured samples of acacia gum collected from trees on Ernetta Island gave very similar chromatograms to each other, and aligned with analyses of modern samples of *Acacia nilotica* from Sudan reported in the literature (Figure 5 [20,30]). In the blackened gums, a compound identified as an inositol, possibly D-pinitol, was observed as the major component (Figures 6 and 7). D-pinitol has been previously identified in *Acacia nilotica* [36], and probably indicates degradation by bacterial attack [37]. A study of black mummification balms, that identified inositols as one of the components, concluded that vegetable tannins were used [38]; another possibility is the presence of degraded gum.

Fucose is not found in *Acacia* gums or fruit gums, but is reported for tragacanth; mannose is found in fruit gums but not in *Acacia* or tragacanth [18,37], although one study reported mannose in tragacanth gum [29]. The presence of fucose and mannose in several samples from both Amara West and Sai therefore suggests that a mixture of gums was used. A similar result was reported for paint samples from one New Kingdom object (17, mummy mask) and two Third Intermediate Period objects (21, mummy mask; 22, falcon) from the Museum of Fine Arts, Boston [39]. The authors concluded that the binders may have included tragacanth but were probably a mixture of gums.

There is a significant presence of glucose in the ancient samples. Plant gums do not typically contain glucose [33,35], although a small amount was detected in the tragacanth reference sample and has been reported by other studies [37]. Other analyses of paint from ancient Egyptian objects, of similar date to the samples studied here, have also detected glucose, which has been interpreted as implying the presence of a sucrose source such as plant juice or honey mixed with the gum [6,35,40]. However, glucose can also originate from degraded cellulosic matter, which may be derived from ancient inclusions of plant material (such as fibres or associated wood) but could also be introduced from modern packaging or labelling materials.

The analysis of local acacia gum indicated that the monosaccharides from these trees are consistent amongst themselves, and blackened (aged) samples mainly consisted of an inositol. This supports the conclusion that the binding media gums from Sai and Amara West were not composed exclusively of acacia gum, and may have been procured from a range of local sources, the full range of which is unknown, or imported from further away. A much larger range of data on ancient sources of gum would be required to draw firm conclusions. Acacia was available locally, and has been identified in the archaeological remains at both sites, as have two kinds of fruit tree, sycamore fig (*Ficus sycomorus*) and doum palm (*Hyphaene thebaica*), which could also have provided sugary substances potentially used as binders [24,25]. Tragacanth (*Astragalus* sp.) is thought to have grown in Turkey and the east coast of the Mediterranean [35]; pistacia resin and bitumen were imported from the same area [41]. Both bitumen and pistacia resin—materials used as pigment and varnish, respectively—have been identified at Sai and Amara West [9,13], along with ceramics from the eastern Mediterranean and Levant [42,43]. Trading networks that would allow the supply of binding materials from the eastern Mediterranean were clearly extant. Polysaccharide gums have been identified from elite funerary or religious contexts in ancient Egypt: on tomb and temple walls, coffins, and mummy masks, from the 18th Dynasty onwards [4,6,17,35,44–47]. However, comparable datasets from the indigenous Kerma culture urban and funerary sites, or from rural and desert sites in the hinterland of

the major pharaonic settlements, are currently lacking. Only with such datasets will it be possible to develop a more nuanced understanding of the movement of materials and any divergences within and between settlements, or how the Egyptian colonial era witnessed the introduction and influence—or not—of painting practices from Egypt itself.

The identification of plant gum in the paint samples from Sai and Amara West demonstrates the use of gum binding media in paints, which may have been destined for projects such as temple decoration, the production of objects in workshops (e.g., samples from E13.29 and E13.31 at Amara West) and for decorated or whitewashed house walls and features. This use of plant gum is consistent with evidence from contemporaneous Egypt itself, and shows that the method for binding pigment into paint was common across types and scales of usage, from elite funerary objects to vernacular domestic architecture. No differences could be distinguished for pigment colour, location on site, or between the sites, and any differences noted may be due to differences in preservation rather than usage.

## 6. Conclusions

The study of securely provenanced samples from modern excavations at two pharaonic sites in Nubia has provided the first analytical evidence for the source of binding media in the paint used to decorate vernacular architecture from the Nile Valley, and the first report of binding media from any site in Nubia (northern Sudan). Furthermore, the identification of monosaccharides in several of the paint samples confirms that a plant gum was being used as a paint and plaster binder within the pharaonic towns of Upper Nubia during the New Kingdom occupation of the mid- to late second millennium BC. The presence in the ancient samples of monosaccharides usually found in different types of gum (fruit, tragacanth, *Acacia*) further suggests that a mixture of gums may have been used, some of which may have been imported from the eastern Mediterranean. This concurs with analyses from Egypt, albeit mostly of elite funerary objects, and suggests that plant gum binders were being used in a similar way in contemporaneous Egypt and Nubia, in both cases availing of materials imported from further afield. Multiple samples of gum taken from locally growing *Acacia nilotica* trees gave extremely similar chromatograms that were not a match for the ancient samples. Naturally blackened samples of the *Acacia nilotica* gum mainly consisted of inositol, which has also been identified in black mummy balm, possibly indicating the use of blackened gum in the balm.

**Author Contributions:** Conceptualization, K.F. and R.J.S.; methodology, K.F. and R.J.S.; formal analysis, K.F.; investigation, K.F.; resources, K.F., R.J.S., N.S. and J.B.; data curation, K.F., N.S. and J.B.; writing—original draft preparation, K.F.; writing—review and editing, K.F., R.J.S., N.S. and J.B.; visualization, K.F.; project administration, N.S. and J.B.; funding acquisition, N.S., J.B. and K.F. All authors have read and agreed to the published version of the manuscript.

**Funding:** Excavations at Sai were funded by the European Research Council project AcrossBorders from 2013 to 2017, ERC grant agreement no. 313668. British Museum Fieldwork at Amara West was funded by the Leverhulme Trust (grant F/00 052/C), British Academy (SG51563) and Qatar-Sudan Archaeological Project (QSAP A.007) from 2008 to 2019. The research on samples from these sites was conducted as part of an Arts & Humanities Research Council collaborative PhD between UCL and the British Museum (Grant 1350956). The project benefited from Wellcome Trust Grant 097365/Z/11/Z supporting the British Museum Department of Scientific Research and RJS salary.

**Institutional Review Board Statement:** Not applicable.

**Informed Consent Statement:** Not applicable.

**Data Availability Statement:** Data is presented in the article.

**Acknowledgments:** The National Corporation for Antiquities and Museums in Sudan generously permitted the export of archaeological samples from both sites for analysis, which provided the opportunity for this study. Antony Simpson is thanked for assistance with maps and figures. Many thanks to anonymous reviewers who were instrumental in improving the text.

**Conflicts of Interest:** The authors declare no conflict of interest.

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
