# Peer review of "Polysaccharide Paint Binding Media at Two Pharaonic Settlements in Nubia"

_heritage, doi:10.3390/heritage5030106_

Round 1
Reviewer 1 Report
this work is well constructed
but does not have innovative character
authos should increase and check bibliographic references
Author Response
Thank you so much for taking time to review our paper. Something happened at the time of upload that removed a lot of the references, for which we apologise. This has now been corrected.
We believe the innovative element of the paper is twofold: (a) binding media for vernacular architecture from the Nile Valley has not previously been analysed and (b) binding media from Nubia has not previously been reported. We have added to the abstract and conclusion to make this clearer.
Reviewer 2 Report
This paper presented a compelling argument for the presence of mixed polysaccharides as paint binders in the paint palettes collected from two sites in Upper Nubia. The work builds on an existing body of work by the authors which identified the colorants, bitumen and Pistacia resin in these samples as, well as polysaccharide binders in other objects of similar age from ancient Egypt. Most interesting is that these palettes were for vernacular rather than funerary use establishing the similarity of materials used by the Egyptians for different functions. The hypothesis that the polysaccharide component is likely to be from the local Acacia nilotica is presented in a balanced manner considering the numerous possibilities from the lively Egyptian trade practices. The source for the sugar additive is also well presented. Polysaccharide analysis by GC/MS and PyGC/MS is a frustrating endeavor and the authors did their best by including analysis of local Acacia nilotica resin collected recently as well as an array of reference materials from the British Museum collection. There are missing references on pages 5, 7, and 12. Also, the spectra are shifted in Figure 4 so that the retention times of the components of AW435 and Sali154 do not line up, and in Figure 5 the components from T1 (Eratta) is also shifted as compared to AW435. How do the authors explain this shift? Two different GC/MS instruments were used for AW435 and Sali154 but the GC method was the same. Furthermore, both AW435 and T1(Eratta) were run on the same instrument with the same GC method. In the end it is not a severe fault as the retention time order remains the same for all samples, its just odd and a short explanation by the authors would remove any concerns.
Author Response
Thank you so much for taking time to review our paper. Something happened at the time of upload that removed a lot of the references, for which we apologise. This has been corrected. Retention times do not quite match because, as you point out, different instruments were used and this does always result in minor differences in RT. AW435 and T1 were run 2 years apart, and instruments change over time, especially since they were all switched off for the majority of 2020. Comments have been added to the text to explain.
Reviewer 3 Report
Dear Authors
I have read with great interest your work that gives new insight into archeological binding media and strategies for their detection. The necessity to match archeological samples with consistent references is always of great interest.
In the text, there are some minor typos and references missing that need to be revised.
Best

Author Response
Thank you so much for taking time to review our paper, and thank you for your comments. Something happened at the time of upload that removed a lot of the references, for which we apologise. This has been corrected. It also seems Figure 1 went awry, that will also be corrected. Comments in the text have been dealt with, many thanks. The formatting will be substantially changed when the text is fitted into the Heritage template, including the tables.
Round 2
Reviewer 1 Report
this work i interesting
it does not present innovative research